# Hypoxia in Aging and Aging-Related Diseases: Mechanism and Therapeutic Strategies

**DOI:** 10.3390/ijms23158165

**Published:** 2022-07-25

**Authors:** Yaqin Wei, Sergio Giunta, Shijin Xia

**Affiliations:** 1Department of Geriatrics, Shanghai Institute of Geriatrics, Huadong Hospital, Fudan University, Shanghai 200000, China; 21211280007@m.fudan.edu.cn; 2Casa di Cura Prof. Nobili–GHC Garofalo Health Care, 40035 Bologna, Italy; sergiogiunta@gmail.com

**Keywords:** hypoxia, aging, aging-related diseases, oxidative stress, therapeutic strategies

## Abstract

As the global aging process continues to lengthen, aging-related diseases (e.g., chronic obstructive pulmonary disease (COPD), heart failure) continue to plague the elderly population. Aging is a complex biological process involving multiple tissues and organs and is involved in the development and progression of multiple aging-related diseases. At the same time, some of these aging-related diseases are often accompanied by hypoxia, chronic inflammation, oxidative stress, and the increased secretion of the senescence-associated secretory phenotype (SASP). Hypoxia seems to play an important role in the process of inflammation and aging, but is often neglected in advanced clinical research studies. Therefore, we have attempted to elucidate the role played by different degrees and types of hypoxia in aging and aging-related diseases and their possible pathways, and propose rational treatment options based on such mechanisms for reference.

## 1. Introduction

In 2019, three scientists—Peter J. Ratcliffe, Willian G. Kealin Jr., and Gregg L. Semenza—were awarded the Nobel Prize in Physiology or Medicine for their outstanding contributions to the field of cellular perception and adaptation to hypoxia, which helped to reveal the molecular mechanisms of oxygen sensing and cellular adaptability to oxygen availability, a field of the link between physiology and medicine (translational physiology). Key molecules involved in hypoxia signaling, known as hypoxia-inducible factors (HIFs), all revolve around the role of the hypoxia-inducible factor-1 (HIF-1) transcription factor at the center of the hypoxia transcription response. HIF is a transcription factor that orchestrates oxygen homeostasis.

HIF-1 is composed of hypoxia-inducible factor-1α (HIF-1α) and hypoxia-inducible factor-1β (HIF-1β) subunits, two subunits that can heterodimerize and bind to DNA. The oxygen-sensing mechanism also involves the tumor-suppressor protein VHL (Von Hippel–Lindau), and HIF-2α [1,2]. In hypoxia, HIF-1α escapes proteasomal degradation and activates the transcription of genes that enable adaptation to reduced oxygen availability, including aspects of the system’s physiology that optimize oxygen delivery. A growing body of research shows that hypoxia is involved in the development of diseases such as tumors, infections, local ischemia, and inflammation. Most mammals in nature rely on oxygen (O_2_) to provide energy for a range of life activities. For example, HIF-1α is an important transcriptional regulator of cellular responses to hypoxia, oxidants, and inflammation, and is overexpressed in the lungs of COPD patients [3].

O_2_ plays a key role in aerobic respiration and cellular metabolism as the final electron acceptor in the mitochondrial electron transport chain. The oxygen content in the body or local tissues will decline (called hypoxia) when the body is unable to meet its metabolic needs due to various pathological factors, which may lead to metabolic crisis and physiological dysfunction. The factors that trigger hypoxia are complex and diverse. According to the duration and characteristics, tissue hypoxia is mainly classified as intermittent hypoxia (HI) or chronic hypoxia, in addition to exercise-induced desaturation (EID). Hypoxia is a potential risk factor for the development of many diseases, especially vascular diseases. In addition, hypoxia can exacerbate the [H^+^] content in tumor microenvironments through the lactic acid pathway, leading to adverse effects such as acidosis and aggravated hypoxia. Intermittent hypoxia is a more common symptom of respiratory diseases and is more common in chronic lung diseases [4]. It is well known that IH is considered to be one of the causative factors of OSA, and recent experimental and clinical studies have shown that obstructive apnea syndrome (OSA) and several respiratory diseases are closely related [5,6,7]. However, there are still some questions that deserve to be explored, such as whether aging is involved in the interaction between OSA and COPD as the prevalence of OSA and respiratory diseases increases with age. The causes of chronic hypoxia are complex and varied. Oxygen levels are lower in the highlands. Therefore, chronic persistent alveolar hypoxia in the population may be responsible for a significant decrease in arterial oxygen pressure (PaO_2_) and oxygen saturation (SaO_2_) compared to the population at low latitudes, and hemodynamic findings indicate enhanced hypoxic pulmonary vasoconstriction [8]. Chronic hypoxia often occurs in the elderly population due to hypoxic ischemic injury, where the microenvironment of cells and the physiological behavior of the cells themselves undergo a number of adaptive responses. However, due to the complex pathological environment of an organism, such physiological changes cannot avoid adverse effects on the organism, such as reactive oxygen species (ROS) aggregation, DNA damage, mitochondrial oxidative stress, inflammation, ferroptosis, etc., leading to aging-related diseases such as cerebral ischemic diseases, cardiovascular diseases, and pulmonary vascular diseases [9,10,11,12]. EID, on the other hand, is mostly found in the group of patients with COPD, IPF, and other lung diseases. The most common definition of EID is a difference between the resting and minimum saturation of peripheral oxygen (∆SpO2) of ≥4% and a minimum saturation of peripheral oxygen (SpO_2_ min) of ≤88% during a 6MWT [13,14,15]. Moreover, several studies have shown that EID may increase the risk of respiratory distress and death in elderly patients [16,17]. Thus, we can find the following common features of aging-related diseases (hypoxia, aging, oxidative stress, inflammation, and major manifestations): (1) hypoxia stimulates the initiation of hypoxia-related signaling pathways and crosstalk senescence signaling pathways downstream, promoting the formation of SASP and other cytokines; (2) local inflammatory factor aggregation in the inflamm-aging state leads to overt inflammation and even triggers a cytokine storm and systemic inflammatory response syndrome (SIRS) [18,19]; (3) almost all chronic inflammatory tissue microenvironments enter into a hypoxic state, and immune system senescence, especially T-cell senescence, promotes the acquisition of senescent phenotypes; (4) vascular lesions and vascular senescence are characterized by the impaired function of various cells (e.g., epithelial cells, vascular smooth muscle cells); (5) progressive accumulation of senescent cardiomyocytes and alveolar cells occurs, and frequent oxidative stress injury occurs under the influence of cellular senescence.

Here, we discuss the effects of hypoxia and oxidative stress on physiological and biochemical activities, and review the potential link between the direct or indirect involvement of hypoxia in the aging process and the role of age-related diseases. As a result, we propose that “hypoxia-aging” plays a central role in aging-related diseases, which we refer to as “hypoxia-aging–aging-related diseases”. Based on our previous research and understanding of antioxidants, we hope to combine antioxidants with anti-aging drugs to provide new therapeutic options for more elderly patients with conditions characterized by hypoxia.

## 2. The Implications of Hypoxia

### 2.1. Oxidative Stress and Mitochondrial Dysfunction

Hypoxic environments favor the accumulation of ROS and increased oxidative stress. In various aging-related diseases and aging processes, hypoxia often leads to a decline in all physiological functions of the body by impairing oxidative phosphorylation and the respiratory chain. However, when hypoxia/ischemia is severe, ROS accumulate and the cellular antioxidant scavenging capacity is not sufficient enough to scavenge the excess ROS; the oxidation–antioxidant imbalance will cause cellular DNA damage and mitochondrial lipid peroxidation, ultimately leading to oxidative damage. A mitochondrial dysfunction. O_2_ completes the tricarboxylic acid cycle (TCA cycle) in the mitochondria, producing energy to support basic physiological activities under normoxic conditions. Therefore, about 70% of ROS are produced in the mitochondria, which can also be considered a self-protective mechanism for cells to concentrate ROS in the mitochondria and thus prevent other organelles from experiencing oxidative stress damage. However, if an organism suffers excessive oxidative damage, it may lead to cumulative damage to the DNA and proteins, which may trigger aging. The current classical aging theory is the “free radical theory of aging” and “mitochondrial theory of aging”, which were successively proposed by Denham Harman [20,21]. The free radical theory of aging suggests that the production of ROS in cells can threaten normal cells and affect human life span. Maintaining appropriate levels of antioxidants and free radical scavengers in the body can help extend the lifespan and delay aging, and a free radical scavenging system consisting of enzymatic and non-enzymatic reactions is present in the organism to protect the body from free radical damage. However, most of the ROS are produced in the mitochondria, and hypoxia leads to the incomplete production of water from O_2_ in the respiratory chain, inducing cells to produce large amounts of ROS. This causes an imbalance in the free radical scavenging system, damaging DNA, proteins, lipids, and biofilms and interfering with normal cellular metabolic activities [22,23]. At the same time, mitochondria themselves are a major target of hypoxic damage [24]. With the continuous revision of this theory, the free radical theory of aging is now preferred as a component of adaptive homeostasis [25]. Several pieces of evidence from animal experiments and clinical trials in recent years suggest that mitochondrial dysfunction and the oxidative damage caused by their ROS production may contribute to age-related disease phenotypes and aging. Mitochondrial abnormalities are found in the epithelial cells and airway smooth muscle cells of COPD patients [26]. Fratta Pasini AM et al. showed that patients with COPD had significantly higher systemic inflammatory markers, such as high-sensitivity C-reactive proteins (hs-CRPs) and leukocytes, and decreased antioxidant glutathione (GSH) compared to patients with no COPD [27]. Moreover, excess ROS triggers ferroptosis, a type of cellular death caused by a reduced cellular antioxidant capacity and ROS accumulation. Experts in the field of ferroptosis research agree that dead cells are accompanied by massive iron accumulation, along with lipid peroxidation, reduced mitochondrial structure, somatic membrane wrinkling, and outer membrane fragmentation. Recent studies suggest that ROS in mitochondria may be involved in regulating immune responses and autophagy-related signaling pathways (e.g., AMPK-ULK1 axis) that trigger iron autophagy [28], and that elevated free iron levels trigger ROS, leading to cellular damage and even death [29].

ROS aggregation undoubtedly poses an oxidative threat to cells. At the same time, ROS can activate the expression of redox-sensitive pro-inflammatory transcription factors to maintain a chronic pro-inflammatory state, thereby increasing the production of cell-derived ROS and creating a vicious cycle [30]. In addition, mitochondrial damage increases with age, leading to the dysregulation of mitochondria-related signaling pathways. Thus, hypoxic environments lead to significant imbalances in mitochondrial homeostasis and cellular energy following hypoxic–ischemic injury in the elderly relative to healthy adults, with adverse effects on cellular metabolism.

### 2.2. Hypoxia-Induced Inflammation

Hypoxia can lead to a range of adaptive responses in tissues and cells, such as increased erythropoietin (EPO) and vasoconstriction [31,32,33]. The research on hypoxia and inflammation is also constantly updated and evolving. Hypoxia-induced inflammation has been confirmed in clinical aspects. Researchers from the University of Edinburgh found hypoxemia and monocytopenia in patients with acute respiratory distress syndrome (ARDS) in the first 48 h after ventilation. Monocytopenia has also been observed in mouse models of hypoxic acute lung injury, which leads to a reduction in monocyte-derived macrophages and enhanced neutrophil-mediated inflammation in the lungs [34]. In the context of respiratory viral diseases, such as influenza and COVID-19 patients, researchers have found that patients with these lower respiratory viral infections often develop hypoxemia, primarily because crosstalk between lung inflammation and a hypoxic microenvironment may impair pulmonary ventilation [35]. Hypoxia may also promote tissue inflammation–fibrosis [36]. In active nodular diseases, hypoxia promotes fibrosis through a pro-fibrotic cytokine response. Hypoxia can reduce antigen presentation and decrease T-cell responses by increasing the pro-inflammatory response of macrophages, which is one of the key factors in nodular diseases. Although the link between hypoxia and inflammation is strong, the complete mechanism has not been elucidated. At the cellular molecular level, studies on hypoxic signaling pathways have shown a close link between hypoxia and inflammation. HIF-1α is a nuclear protein with stable transcriptional activity, which is stably expressed in response to hypoxic stimulation and can enter the nucleus to regulate the expression of downstream target genes. HIF-1α regulates inflammation and oxidative stress following hyperglycemia and hypoxia-induced vascular cell injury [37]. The induction of HIF expression under normoxia may contribute to diseases with a chronic inflammatory component. At the same time, chronic low-grade inflammation (inflamm-aging) conditions the abnormal activation of transcription factors, which leads to changes in the balance of growth factors, chemokines, cytokines, and ROS, resulting in inflammation and cancer [38]. Therefore, the rational regulation of HIF is beneficial for reducing inflammatory responses after disease injury [39]. In addition, inflammatory factors and microbial infections activate immune cells and induce glycolysis to occur, as well as activate HIF-1α [40]. The metabolites of HIF-1α and glycolytic pathways can promote inflammation during inflammatory activation and the migration of macrophages and dendritic cells [41,42,43]. Hypoxia-inducible factors promote myeloid macrophage metabolism. Macrophage HIF activation contributes to the orchestration of tissue repair responses. A study using a mouse model of myocardial infarction and a mouse model with conditional genetic loss and gain of function showed that both HIF-1α and HIF-2α had unique pro-inflammatory effects. Among them, HIF1α protected cardiomyocytes through macrophage glycolysis reprogramming. HIF-2α had the function of inhibiting mitochondrial metabolism in anti-inflammatory macrophages [41]. Watanabe S et al. directly showed that hypoxia induced NLRP3 inflammatory vesicle activation in macrophages [44]. An O_2_ percentage of 0.1% induced mouse macrophages to promote interleukin 1β (IL-1β) and caspase-1 synthesis while releasing IL-1β and lactate dehydrogenase, suggesting that hypoxia induces NLRP3 inflammatory vesicle drive, leading to inflammation. In vitro and in vivo experiments demonstrated that NLRP3 deficiency and a specific caspase-1 blockade prevented hypoxia-induced IL-1β production and release. Thus, NLRP3 inflammatory vesicles can sense early warning signals of intracellular energy crises and hypoxia. In the nervous system, when mouse neurons and microglia were exposed to hypoxia, S100A8 protein (a pro-inflammatory mediator) expression was increased, which induced neuroinflammation and neuronal apoptosis. The secretion of S100A8 by neurons under hypoxic conditions activated the secretion of tumor necrosis factor-α (TNF-α) and interleukin 6 (IL-6) by phosphorylating the microglia c-Jun terminal kinase (JNK). Meanwhile, the phosphorylation of the extrinsic signal-regulated kinase (ERK) via the TLR4 receptor induced the initiation of NLRP3 inflammatory vesicles [45]. In addition, intermittent hypoxia in the microenvironment of tumor solid tissues promoted a pro-inflammatory phenotype characterized by an increased secretion of cytokines, such as TNF-α and interleukin 8 (IL-8), by macrophages through the JNK/p65 signaling pathway [46]. Moreover, intermittent hypoxia significantly increased the activation of c-Jun and NF-κB in M1-type macrophages, and the inhibition of c-Jun and p65 activation demonstrated the involvement of p65 in the pro-inflammatory effects of hypoxia on macrophages. The above studies gradually revealed the close connection between hypoxia and inflammation from the study of hypoxic signaling pathways, and also observed the adverse consequences of inflammatory responses in hypoxic environments in clinical settings, especially considering that local hypoxia caused by hypoxemia and its stimulation of immune responses are often overlooked. Therefore, an in-depth study on and elucidation of the relationship between both NF-κB and HIF would greatly enhance drug development.

## 3. Hypoxia-Induced Aging

### 3.1. Characteristics of Aging

Aging is a complex biological network process, and a major feature of senescent organisms is the accumulation of senescent cells. Cellular senescence is mainly reflected in replicative and stress-related senescence, such as the activation of p53 and p16INK4a, respectively, leading to p21CIP1 activation and permanent cell cycle arrest, which promotes tissue remodeling during development and after injury, but may also lead to a decrease in tissue regeneration potential and function, resulting in inflammation and the development of various chronic diseases or tumors in senescent organisms. The mechanisms of aging that have been proposed include the free radical senescence theory, telomere shortening senescence theory, metabolic waste accumulation theory, genetic senescence theory, immune senescence theory, mitochondrial damage senescence theory, protein cross-linked senescence theory, divine trip endocrine senescence theory, and the latest additions of autophagic senescence theory [47], inflamm-aging theory [48], and stem cell senescence theory [49]. Among them, telomere shortening and mitochondrial damage also trigger inflammation and oxidative damage, suggesting that aging seems to be a complex traffic junction—an important juncture associated with multiple pathophysiological activities. 

One of the fundamental features of cellular senescence, which is the basis of current studies on the mechanisms of aging is the appearance of an SASP, which represents inflammation and tissue remodeling [50]. Existing studies have confirmed the aberrant secretory function of senescent cells, with the presence of a large number of cytokines, growth factors, chemokines, matrix metalloproteinases, and lipid molecules among the secreted proteins, and their biological relevance in influencing the metabolic activity of local and surrounding stromal cells in order to maintain the homeostasis of the cellular survival environment [51]. SASP does not usually refer to a single pro-inflammatory substance and its function, as the combination and role of the secreted substances produced varies according to different aging triggers. This may be related to the duration of the role of SASP, with a recent study showing that primary mouse keratin-forming cells briefly exposed to SASP exhibited increased stem cell markers and an increased regenerative capacity in vivo, but that a prolonged exposure to SASP led to subsequent cellular senescence arrest and interference with normal cellular physiological activities [52]. Although SASP can promote tissue repair functions [53], when the SASP regulatory system is imbalanced, pro-inflammatory factors including IL-6, IL-8, membrane cofactor proteins, and macrophage inflammatory proteins exert deleterious effects in both autocrine and paracrine ways [54], such as promoting cell proliferation, angiogenesis, and inflammatory responses [55,56]. In addition, since most SASP factors are pro-inflammatory, the increase in senescent cells with age may lead to low levels of chronic inflammation, which is closely associated with aging and age-related diseases [57]. Oxidative stress and persistent DNA damage are also associated with SASP [58]. Different aging-inducing stimuli may activate stress-inducible kinases in fibroblasts through the accumulation of damage to the DNA p38MAPK, which induces SASP production mainly by increasing NF-κB transcriptional activity. The inhibition of p38MAPK significantly reduces the secretion of most of the SASP factors and also attenuates the paracrine effects of senescent cells [59]. Therefore, the rational regulation of SASP in senescent mammals is particularly important, although there is no clear means of regulation and the corresponding regulatory effects. Cellular senescence is often accompanied by autophagy. Autophagy is a very complex and highly conserved biological process, and there are three main types: macroautophagy, microautophagy, and chaperone-mediated autophagy (CMA). The prevailing view is that autophagy is a highly selective cellular clearance pathway associated with the maintenance of homeostasis, contributing to the timely removal of senescent cells while reducing the accumulation of inflammatory factors. Selective autophagy is divided into many subtypes, including mitochondrial autophagy: the binding of damaged mitochondria to mitophagy receptors, leading to mitochondrial engulfment and subsequent degradation by lysosomes. As mentioned above, hypoxia-induced ROS aggregation leads to mitochondrial damage. Mitochondrial autophagy is a pivotal method for eliminating ROS, while hypoxia is at the same time one of the important factors involved in inducing mitochondrial autophagy [60], which can be seen as a self-regulatory and protective method for organisms to maintain the balance of their internal environment homeostasis. Several clinical studies have shown that hypoxia-induced mitochondrial autophagy may benefit patients with ischemia-reperfusion injury [61,62]. This seems to imply the positive significance of hypoxia-induced mitochondrial autophagy. However, in most mammals, aging leads to impaired autophagy [47] and undermines the protective effect of mitochondrial autophagy. We suggest that this is likely to disrupt the balance between the hypoxia–ROS–mitochondrial autophagy triad, which, together with crosstalk in the signaling pathway between hypoxia, mitochondrial autophagy, and aging, ultimately leads to mitochondrial autophagy, promoting the development of age-related diseases [63].

### 3.2. Hypoxia Causes Aging by Inducing Changes in Specific Genes and Signaling Pathways

When tissues are subjected to acute injury resulting in ischemia/hypoxia, cells adapt to the hypoxic environment by inducing the expression of a number of adaptive genes and regulating post-translational modifications. These adaptive changes in tissue cells in hypoxic environments are controlled by the HIF family. The dysregulation or overexpression of HIF-1α induced by hypoxia is associated with many pathological processes, such as cardiovascular diseases, metabolic diseases, and tumors. For example, in lung diseases, HIF-1α induces the expression of the vascular endothelial growth factor, ROS, and inducible nitric oxide synthase (iNOS) through multiple signaling pathways and a broad target gene profile, promoting an increased inflammatory response. This leads to endothelial cell dysfunction and leukocyte adhesion, promoting the proliferation of pulmonary artery smooth muscle cells (PASMCs) and O_2_ delivery to hypoxic regions [64,65,66,67]. At the same time, senescence may also be involved in promoting the expression of HIF-1α [68] (see Figure 1). During hypoxia and aging, the hypoxic signaling pathway interacts with the sirtuin, AMPK, and NF-κB signaling pathways. For example, hypoxia induces an inflammatory response in cells, and the activation of the NF-κB pathway in endothelial cells facilitates the release of cellular inflammatory factors and acts as positive feedback for HIF-1. There is an interconnection between HIF and the sirtuin family. SIRT1 and HIF-1α jointly regulate mitochondrial senescence, and SIRT1 has a regulatory effect on HIF-1α activity; however, the specific regulatory mechanism has been controversial. The evidence [69,70,71] has shown that SIRT1 deletion or inactivation under hypoxic conditions leads to reduced hypoxic HIF-1α accumulation, accompanied by increased HIF-1α acetylation, that SIRT1 assists in stabilizing the HIF-1α protein through direct binding and deacetylation, and that the upregulation of SIRT1 may prevent premature cellular senescence and the development of many chronic diseases associated with aging. AMPK is an important regulator of energy metabolism, resilience, and cellular proteostasis, and hypoxia can activate AMPK directly or indirectly. However, the activation capacity of AMPK signaling decreases with age, which impairs the maintenance of cellular homeostasis and accelerates the aging process, thus triggering a variety of aging-related diseases [72,73].

## 4. Hypoxia-Mediated Cellular Senescence and Inflammatory Responses Are Involved in the Onset and Development of Aging-Related Diseases

### 4.1. COPD

COPD is a common respiratory disease in the elderly. Its main pathophysiological characteristic change is the long-term chronic inflammation of the lungs due to persistent airflow restriction, resulting in pulmonary ventilation dysfunction and a hypoventilated state. COPD can be triggered by smoking, oxidative stress, etc. The acute exacerbation of COPD occurs mainly after the infection and inflammation of the respiratory tract, which predicts aggravation of the patient’s condition and leads to a decrease in quality of life or even death.

Clinically, elevated HIF-1α levels have been observed to be associated with disease progression in COPD [74]. The more commonly accepted view is that tobacco bacteria stimulate elevated inflammatory factors such as respiratory NF- κB and interleukin 1 (IL-1), leading to oxidative stress in airway epithelial cells, and the resulting ROS activate PI3K/AKT/mTOR signaling pathways that promote the upregulation of HIF-1α gene expression and the production of vascular endothelial growth factor (VEGF), basic fibroblast growth factor-2 (FGF-2), platelet-derived growth factor (PDGF), and other factors, thereby inducing an inflammatory response [75,76] (see Figure 2). Elevated HIF-1α upregulates platelet-activating factor receptor (PAFR) expression [77], and as a major adhesion receptor for respiratory bacteria, PAFR can promote airway inflammation and vascular endothelial cell injury when upregulated [78]. Therefore, clinical studies should pay attention to PAFR indicators in COPD patients, which may be a red flag for recalcitrant bacterial infections or a predisposition to pneumonia [79].

The effect of hypoxia on coagulation in COPD patients cannot be ignored. Hypoxic stimulation given to COPD patients for 2 h leads to coagulation activation and the elevation of factors such as thrombin-antithrombin complexes (TAT), D-dimers, and the von Willebrand factor antigen (VWF:Ag), while inflammatory factors have also been observed to be elevated [80]. Obesity may advance and exacerbate the hypoxic crisis of COPD. I. García-Talavera et al. collected data on respiratory function tests, cardiovascular comorbidities, body mass index, and 6-minute walk test results in a cross-sectional retrospective study. Patients who desaturated during the first minute of the test were referred to as early oxygen desaturators. Their study indicated that obesity might behave as a marker for early oxygen desaturation. This simple measure might indicate potential early oxygen desaturation in COPD patients [81]. Recent studies point to the involvement of lung epithelial cell senescence in the development of COPD [82]. The direct evidence for this is the increased secretion of SASP from alveolar lavage fluid in COPD patients. Smoke-induced senescent cells interfere with lung function by regulating miRNAs and are involved in lung epithelial cell senescence via Sp1/SIRT1/HIF-1α. This not only enriches the pathogenesis of COPD, but also facilitates the development of a more coherent treatment plan, taking into account hypoxia and senescence.

### 4.2. Pulmonary Hypertension (PH)

PH is a serious pulmonary vascular disease that can cause death in up to 2/3 of patients within 5 years of diagnosis. The current estimates suggest a prevalence of pulmonary hypertension in approximately 1% of the global population, with this prevalence reaching 10% in people over 65 years of age [83]. According to the latest WHO-recommended clinical classification of PH, hypoxia-induced PH is classified in group III [84].

More than 140 million people worldwide live at altitudes above 2500 m. High-altitude pulmonary hypertension (HAPH) affects 5% to 10% of the population, with symptoms such as dyspnea and decreased exercise tolerance. The oxygen saturation (SaO_2_) of the partial pressure of oxygen in the arteries (PaO_2_) is lower in the high-altitude population than in the low-altitude population [85,86]. The pathophysiological mechanisms are worth discussing: on the one hand, chronic persistent alveolar hypoxia was observed in high-altitude residents. On the other hand, the hemodynamic findings suggest the presence of enhanced hypoxic pulmonary vasoconstriction and pulmonary vascular remodeling. Both lead to a decrease in the lumen diameter and an increase in pulmonary vascular resistance. In vivo studies have shown that chronic persistent hypoxic exposure can lead to a thickening of the pulmonary artery vessel wall [87,88]. If tissue cells are persistently hypoxic, this will promote the excessive proliferation of PASMCs and pulmonary artery endothelial cells (PAECs), the further thickening of small pulmonary arteries, and the formation of irreversible myenteric arteries. Therefore, hypoxia-associated smooth muscle cell proliferation may be one of the pathophysiological mechanisms for the development of HAPH [89,90,91].

A case-control study [92] found that pulmonary hypertension was associated with sleep apnea and hypoxemia in a highland population, and patients were often combined with OSA, resulting in a hypoxic state of the tissues and cells. This suggests a pathophysiological interaction between pulmonary hemodynamics and sleep apnea. OSA typically manifests after repeated sleep apnea hyperventilation, which leads to intermittent hypoxia. Costa-Silva JH et al. demonstrated that intermittent hypoxia can lead to repeated significant increases in pulmonary artery pressure (PAP) in patients with OSA [93]. This is associated with excitation or the instability of sympathetic activity in nocturnal patients, and this repeated intermittent hypoxia may cause inflammatory responses, sympathetic activation, and hypoxia-reoxygenation damage. When tissue cells appear stressed by hypoxia, not only can this cause inflammation through mononuclear macrophage activation, but this can also lead to monocyte migration to the vessel wall—a process that causes intimal damage, thickening, and eventually atherosclerosis or vascular occlusion, further aggravating PH and creating a vicious cycle [94,95,96]. The above study points to a possible potential link between HAPH and sleep apnea, which deserves to be studied and explored in depth.

### 4.3. Cardiovascular Diseases

Cardiovascular diseases are a typical group of aging-related diseases, mainly including myocardial infarction, coronary artery disease, and heart failure. As the global aging process lengthens and the complex mechanisms of aging are investigated, several signaling pathways linking the aging process and cardiovascular diseases have attracted the attention of researchers [97,98,99]. As a result, aging is increasingly studied as an important risk factor for cardiovascular diseases (CVD).

Cardiac aging is a complex process. Senescent individuals are commonly affected by immune senescence and inflamm-aging. As a result, senescent cardiomyocytes may be accompanied by chronic low-grade inflammation. This chronic low-grade inflammation due to senescence promotes the secretion of cellular inflammatory factors such as IL-6 and TNF-α [100,101]. These inflammatory mediators promote vascular senescence and vascular injury (e.g., atherosclerosis) by activating multiple signaling pathways (e.g., the ATM/p53/p21 (WAF1/Cip1) signaling pathway and the NF-κB/TOM signaling pathway) to induce senescence or injury in endothelial cells and vascular smooth muscle cells [102,103]. Inflamm-aging simultaneously enhances platelet reactivity [104]. When a low-grade inflammatory state develops into significant inflammation due to infection or other causes, arterial plaques are at a risk of rupturing, and if patients do not have timely vascularization in the event of plaque rupture, their cardiomyocytes will begin to die due to O_2_ deprivation [105]. In fact, cardiomyocytes are susceptible to myocardial injury from ischemia-reperfusion when acute hypoxia occurs. Adenosine has a certain cardioprotective effect, and in this regard, targeting the hypoxia-adenosine link for controlling excessive inflammation has been proposed to benefit patients from an anti-inflammatory perspective [106]. Therefore, we hope to provide a multifaceted program to prevent inflammation and plaque rupture in advance by using the aspect of controlling inflamm-aging and avoiding hypoxia as much as possible, so as to prolong the stabilization period and improve the quality of life of the patients.

## 5. Antioxidant- and Anti-Aging-Based Therapeutic Perspectives

### 5.1. Antioxidant Therapeutic Strategies

Hypoxia and oxidative stress have become integral components of many aging-related diseases. Endogenous ROS generated by mitochondrial dysfunction are involved in a range of cellular physiological and metabolic activities, and proteins are more susceptible to ROS due to their unique biochemical properties, forming macromolecular complexes that together contribute to aging and aging-related diseases under the complex regulation of oxidative stress-response signaling networks. Therefore, in order to target against oxidative stress damage and scavenge ROS, research on antioxidants is gradually increasing and the findings are increasingly used in the clinic to benefit patients.

#### 5.1.1. Superoxide Dismutases (SODs)

SODs are important members of the antioxidant enzyme family in biological systems and are widely distributed. SODs play a vital role in the oxidative and antioxidant balance of the body as superoxide anion radical scavengers by catalyzing the dismutation of superoxide anion radicals to generate oxygen and hydrogen peroxides, preventing lipid peroxidation, and inhibiting the production of inflammatory cytokines and the inflammatory responses caused by oxidative stress [107]. SOD1 catalyzes the dismutation of ROS to produce H_2_O_2_ and O_2_, and the H_2_O_2_ produced by SODs is eliminated by glutathione peroxidases (GPXs), which mainly play a role in delaying aging and anti-inflammation [108,109]. When oxidative stress occurs in the body, the mouse nuclear hormone transcription factor peroxisome proliferator-activated receptor γ coactivator1-α (PGC1α) protects against self-oxidative damage by inducing SOD1 expression and increasing SOD1 enzyme levels [110]. It has been suggested that SOD may be a potential antioxidant enzyme of PH, regulating the onset and progression of PH [111,112]. With increasing age, SOD gradually decreases and its activity also decreases. Therefore, there is a need to find suitable analogues or substances that can upregulate SOD expression. Recent studies have shown that Momordica saponin extract (MSE) not only up-regulates SOD gene expression, but also increases SOD activity, significantly enhancing the anti-stress and anti-aging activity of *Caenorhabditis elegans* [113]. However, whether MSE can be used in clinical applications still needs further study.

#### 5.1.2. NAD^+^ and NAD^+^ Precursors

NAD^+^ is an antioxidant that can effectively scavenge ROS and regulate the balance of the redox state of the body [114]. As an important mediator of electron transfer, NAD^+^ is widely involved in DNA repair, epigenetic modification, and the regulation of body inflammation, biological rhythms, and other physiological processes; it is also closely related to aging and aging-related diseases [115]. Therefore, scientists are seeking effective means to increase NAD^+^ levels, and NAD^+^ precursors are an important way to supplement NAD^+^. NMN, NR, and niacin are the hot spots of research, among which NMN and niacin have been published in human clinical studies in 2020–2021. The results showed that NMN significantly improved muscle strength, exercise capacity, and hearing, while the blood levels of NAD^+^ and NMN metabolites were significantly elevated [116,117,118]. The anti-inflammatory effects of high doses of niacin were positive, and preliminary clinical trials seem to predict that niacin can significantly increase blood NAD^+^ levels [119,120]. However, current evidence still does not recommend niacin supplementation to increase NAD^+^ levels, as this may increase patients’ blood glucose levels and homocysteine levels [121,122]. Elevated homocysteine levels are clearly associated with cardiovascular disease and age-related diseases such as dementia, while Liu, L suggests that tryptophan may be the real source of the NAD^+^-raising function [123]. In conclusion, the research on NAD^+^ precursors will continue and we hope to find an effective way to supplement NAD^+^ for the purpose of fighting oxidative stress damage and delaying aging.

#### 5.1.3. NADPH Oxidase (NOX) Inhibitors

As the initial ROS-producing enzyme, NOX is the main source of ROS in the blood vessels. It affects the vascular function by dysregulating or uncoupling eNOS, inducing inflammation and leading to vascular endothelial dysfunction [124]. Therefore, more and more NADPH oxidase (NOX) inhibitors have been investigated to scavenge ROS, reduce inflammatory factors, protect blood vessels, and treat vascular-related diseases [125,126,127]. There are seven NOX isoforms with different distribution characteristics and expression levels in different tissues and organs throughout the body. The specific inhibition of different NOX isoforms expressed in different spaces can not only reduce ROS production, but also avoid unnecessary damage, thus compensating for the deficiencies of general antioxidants. For example, phagocytic NOX2 is highly expressed in neutrophils and macrophages. The use of NOX2 inhibitors inhibits the induction of TNF-α, IL-1β, and IL-6 and attenuates tissue inflammation [128]. NOX4 knockdown attenuated lung ROS production in septic mice by reducing the associated redox-sensitive related signaling pathways and endothelial cell dysfunction. Targeting NOX4 attenuates sepsis-induced acute lung injury [129].

#### 5.1.4. GSH and N-Acetyl Cysteine (NAC)

GSH is the most abundant antioxidant in the cells. The high content of glutathione in erythrocytes is important for eliminating the damaging effect of oxidants on the erythrocyte membrane structure and maintaining the stability of the erythrocyte membrane structure. Therefore, when tissue ischemia occurs, the GSH redox system is unable to maintain intracellular GSH/glutathione oxidation (GSSG) homeostasis, which can disrupt normal cellular metabolic processes. This is especially evident in the lungs [130]. Therefore, the timely supplementation of GSH and its precursor NAC, a mucolytic and anti-inflammatory agent, is of positive interest for COPD patients characterized by hypoxia and oxidative stress [131]. The results of a meta-analysis showed symptomatic relief in patients with chronic bronchitis or COPD treated with NAC, with a sustained reduction in exacerbation pathology that was well-tolerated by the patients, with few and non-dose-dependent adverse effects [132].

Whether long-term supplementation with GSH and NAC is encouraged in healthy older adults without co-morbidities is controversial, as the effects of long-term supplementation on longevity are unclear. Long-term administrations of GSH or NAC have been studied to disrupt the overall gene expression found in *Cryptobacterium hidradenum* and accelerate the aging process. In contrast, an avoidance of thiol-based diets appears to be more beneficial for extending lifespan [133].

### 5.2. Anti-Aging Therapy

Aging is a complex biological process in which multiple substances are involved. Therapeutic regimens are developed to target aging-related targets or pathways, including reducing cellular DNA damage from free radicals generated by oxidative stress, modulating immune aging, suppressing inflammatory responses, and scavenging senescent cells. Finding the right biomarkers allows researchers to properly assess and intervene in aging. A groundbreaking study has shown that the metabolic inhibition of glutamine can exert anti-aging effects by specifically eliminating senescent cells and improving organ function [134]. However, the body’s protein requirement increases with age, while amino acids (e.g., leucine, methionine) activate aging pathways such as the mTOR pathway, which is one of the important aging pathways. The idea of utilizing an amino acid-restricted diet against aging has been proposed in response. Studies have shown that methionine transsulfuration levels are upregulated in long-lived populations, and that restricting methionine levels could help improve metabolic health and extend lifespan [135]. The above studies suggest that biomarkers are the basis for anti-aging research, especially pharmaceutical research.

#### 5.2.1. JAK Inhibitor: Ruxolitinib

In senescent cells, the Janus kinase/signal transducer and activator of transcription (JAK/STAT) signaling pathway activity is enhanced. Several studies have shown that JAK inhibitors exert anti-aging functions through the JAK/STAT signaling pathway by regulating various cytokines, including SASP [136,137,138]. Ruxolitinib is a selective JAK1/2 inhibitor that was originally approved by the FDA for the treatment of myelofibrosis and polycythemia vera [139,140]. It was shown that ruxolitinib inhibition of JAK1/2 prevents premature cell death by reducing induced cell cycle arrest, cellular senescence, and nuclear aberrations in mice [141]. The overactivation of AKT/mTOR, a downstream signaling pathway of JAK, leads to interstitial fibrosis, and treatment with ruxolitinib effectively reduces the inflammation and oxidative stress induced by unilateral ureteral obstruction (UUO), probably because ruxolitinib reduces the activity of the AKT/mTOR pathway and reduces SASP damage to renal tubular epithelial cells [142]. Therefore, we hypothesize that JAK inhibitors may help to reduce the vulnerability of an organism by inhibiting SASP. Targeting the JAK pathway to inhibit SASP is expected to treat age-related dysfunction. In addition, rapamycin has been shown to function as a SASP inhibitor. The mTOR signaling pathway regulates metabolism and multiple physiological processes in organisms. The dysregulation of mTOR signaling is associated with a variety of age-related diseases. Rapamycin is an inhibitor of the target of rapamycin complex 1 (mTORC1). The pallamycin analogue DL001 effectively inhibits mTORC1 activity and its signaling pathway, with no significant side effects on glucose metabolism or the immune system [143].

#### 5.2.2. Senolytics

Senolytics are a novel class of drugs that selectively kill senescent cells, including BCL family inhibitors, Navitoclax, Panobinostat, PI3K/AKT inhibitors, dasatinib, quercetin, fisetin, and HSP90 inhibitors. One of the classic anti-aging drug combinations is dasatinib plus quercetin (DQ). Cell culture experiments have shown that dasatinib targets the elimination of senescent human adipocytes, while quercetin was more effective against senescent human endothelial cells and mouse bone marrow mesenchymal stem cells [144]. The results were quickly applied to clinical applications. Researchers reportedly administered dasatinib (100 mg/day) and quercetin (1250 mg/day) to older adults with a clear diagnosis of idiopathic pulmonary fibrosis. The results showed that the “dasatinib + quercetin” treatment was effective in relieving the patients’ frailty, improving their mobility and extending their healthy lifespan [145]. Another clinical study in the same year suggested that after 11 days of DQ in patients with diabetic nephropathy, the expression of p16INK4A by senescent adipocytes was reduced and SASP levels (e.g., IL-6, IL-1α) were also decreased [146]. The intestinal flora changes with the normal aging process in mice. Anti-aging treatments with DQ helped to alleviate intestinal aging, suppressed inflammation, and regulated the intestinal flora in aged mice [147]. The above studies suggest that the selective removal of senescent cells by DQ may be a way to extend healthy lifespans through the targeted anti-aging treatment of various common age-related diseases. In addition, compared with drug-interfered senescent cell clearance techniques, the induced mouse INKT cells had more efficient and safer senescent cell clearance and senescence reversal abilities, and the treatment was able to rapidly clear lung senescent cells in mice with specific pulmonary fibrosis, reduce SASP levels, and improve pulmonary fibrosis in mice [148]. Therefore, whether this result can be applied in the clinic needs to be supported by more future studies.

## 6. Conclusions

Hypoxia, oxidative stress, and inflammation are potential networks of aging and aging-related diseases. Hypoxia drives inflammation and senescence through multiple pathways, and in the absence of intervention, the inflammation and senescence can in turn exacerbate local tissue hypoxia. Immune senescence causes a failure of immune cells to recognize and clear senescent cells in time. Healthy older adults in a state of inflamm-aging may suffer from hypoxia, which leads to massive ROS accumulation and oxidative stress, as well as inflammation, an increased release of inflammatory factors and mediators, the destruction of endothelial cells and vascular smooth muscle cells, and vascular senescence. Based on these theories, we hypothesized that hypoxia acts as an activator of oxidative stress and inflammation, promoting inflammatory aging and vascular senescence aging, and involved in regulating cardiovascular aging and cardiovascular disease. Therefore, targeting antioxidants, removing ROS, and targeting anti-aging may benefit patients. However, there is a lack of useful biomarkers to respond to chronic inflammatory activity and to determine which patients will benefit most from antioxidant therapy or to indicate how many doses are required to restore redox homeostasis in patients. Practical guidance on how to promote the immune cell recognition and clearance of senescent cells, as well as how to reduce or eliminate systemic chronic inflammation, will help fight aging and aging-related diseases.

## Figures and Tables

**Figure 1 ijms-23-08165-f001:**
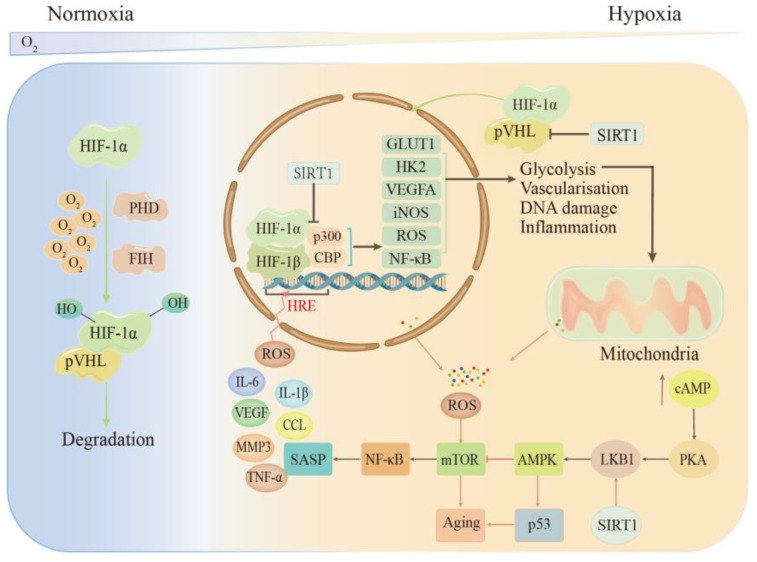
Regulation of HIF-1α under hypoxic conditions. Under normoxia, HIF-1α is inhibited by PHD and FIH and cannot bind to HIF-1β. Hypoxia binds to HIF-1β and recruits p300 or CBP to respond adaptively to hypoxia, inducing the expression of many genes such as GLUT1 and HK2, thus promoting glycolysis to produce energy and also stimulating angiogenesis and DNA damage. Intracellular cAMP levels increase after sustained hypoxia, and cAMP and senescence act through signaling pathways such as AMPK-mTOR, ultimately producing SASP to play a physiological role. PHD—prolyl hydroxylase domain; FIH—factor inhibiting HIF-1; pVHL—VHL (Von Hippel–Lindau) tumor suppressor protein; SASP—senescence-associated secretory phenotype; ROS—reactive oxide species; TNF-α—tumor necrosis factor-α; VEGF—vascular endothelial growth factor; CCL—chemokine; HRE—hypoxia response element; GLUT1—glucose transporter type 1; HK-2—hexokinase 2; iNOS—inducible nitric oxide synthase.

**Figure 2 ijms-23-08165-f002:**
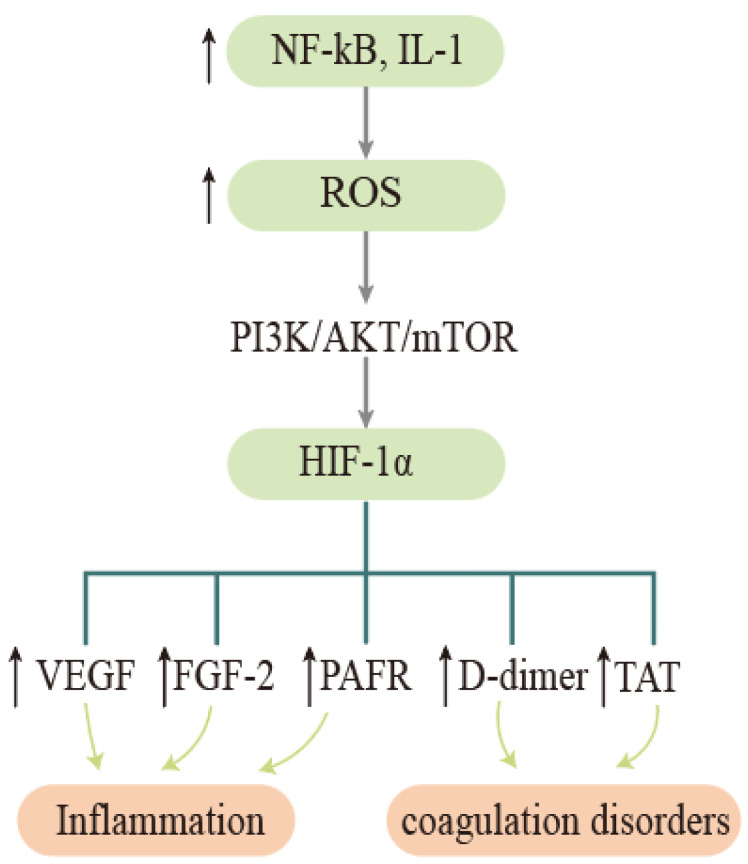
Oxidative stress and hypoxia lead to inflammation and coagulation dysfunction in COPD. Tobacco and bacteria stimulate inflammatory factors such as elevated respiratory NF- κB and IL-1, leading to oxidative stress in airway epithelial cells, which promotes upregulation of HIF-1α gene expression through the PI3K/AKT/mTOR signaling pathway, and HIF-1α entry promotes the expression of factors such as VEGF, FGF-2, and TAT, inducing inflammatory responses and coagulation dysfunction. ROS—reactive oxygen species; VEGF—vascular endothelial growth factor; FGF-2—fibroblast growth factor; PAFR—platelet-activating factor receptor; TAT—thrombin-antithrombin complex.

## Data Availability

Not applicable.

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
