# Peer review of "Hypoxia in Aging and Aging-Related Diseases: Mechanism and Therapeutic Strategies"

_ijms, 2022, doi:10.3390/ijms23158165_

Round 1

Reviewer 1 Report

Wei et al., in their article "Hypoxia in Aging and Aging-Related Diseases: Mechanism and 2 Therapeutic Strategies," examine the function of different degrees and types of hypoxia in aging and aging-related disease pathways, and provide logical therapeutic alternatives based on such processes. The book is beautifully written and discusses hypoxia's function in aging. It covers all current features of hypoxia. The authors performed an outstanding job. There was only one minor issue:

1. The abbreviated form of tricarboxylic acid, TAC, should be changed to TCA on page 3 line 110.

Reviewer 2 Report

This review article sought to elucidate the role played by different degrees and types of hypoxia in aging and aging-related diseases and its possible pathways.  In this review, the authors summarized and discussed the effects of hypoxia and oxidative stress on physiological or biochemical activities, and review the potential link between the direct or indirect involvement of hypoxia in the aging process and the role of age-related diseases.  This manuscript is interesting, but I have some concerns that need to be addressed as follows:

Major concerns:

1.      A growing body of research has shown that mitophagy plays a pivotal role during aging.  Thus, the authors should provide more information about the effects of hypoxia on mitophagy in aging and aging-related diseases.  

2.      For the antioxidant and anti-aging based therapeutic perspectives, recent clinical trials have shown that hyperbaric oxygen therapy (HBOT) can target aging hallmarks, including telomere shortening and senescent cells clearance.  Thus, authors should also include the HBOT in this section.

Minor concerns:

1.      Page 7, Figure1, “Hypoxia” should be moved upward and placed at the same level as “Normoxia”.

2.      Page 7, Line 309, “iInvolved” should be “Involved”.

Round 2

Reviewer 2 Report

The authors have addressed my concerns, and I recommend publication in the journal.